Functional characterization of the GhNRT2.1e gene reveals its significant role in improving nitrogen use efficiency in Gossypium hirsutum

Zhang Xinmiao 1
Feng Jiajia 1
Zhao Ruolin 1
Cheng Hailiang 1
Ashraf Javaria 2
Wang Qiaolian 1
Lv Limin 1
Zhang Youping 1
Song Guoli 1 sglzms@163.com
Zuo Dongyun 1 zdy041@163.com
1 State Key Laboratory of Cotton Biology, Institute of Cotton Research, Chinese Academy of Agricultural Sciences , Anyang, Henan , China
2 Department of Plant Breeding and Genetics, Faculty of Agriculture and Environment, The Islamia University of Bahawalpur , Bahawalpur, Punjab , Pakistan
Kumar Sushil
Electronic publication date: 2023 Mar 28
Publication date: 2023
Volume: 11
Electronic Location ID: e15152
Received 2022 Dec 7; Accepted 2023 Mar 10
Copyright: © 2023 Zhang et al.
Copyright year: 2023
Copyright holder: Zhang et al.
License: This is an open access article distributed under the terms of the Creative Commons Attribution License, which permits unrestricted use, distribution, reproduction and adaptation in any medium and for any purpose provided that it is properly attributed. For attribution, the original author(s), title, publication source (PeerJ) and either DOI or URL of the article must be cited.
License URL: https://creativecommons.org/licenses/by/4.0/

Keywords: NRT2, NUE, Gene family, Expression pattern, High-affinity nitrate transporter, Gossypium hirsutum

Funding: National Natural Science Foundation of China 31621005 and 31901581 Central Public-interest Scientific Institution Basal Research Fund 1610162022054 and 1610162022016 Agricultural Science and Technology Innovation Program of Chinese Academy of Agricultural Sciences This work was supported by the National Natural Science Foundation of China (Grant No. 31621005 and Grant No. 31901581), the Central Public-interest Scientific Institution Basal Research Fund (Grant No. 1610162022054 and Grant No. 1610162022016) and the “Agricultural Science and Technology Innovation Program of Chinese Academy of Agricultural Sciences”. The funders had no role in study design, data collection and analysis, decision to publish, or preparation of the manuscript.

==============================
Background

Nitrate is the primary type of nitrogen available to plants, which is absorbed and transported by nitrate transporter 2 (NRT2) at low nitrate conditions.

Methods

Genome-wide identification of NRT2 genes in G. hirsutum was performed. Gene expression patterns were revealed using RNA-seq and qRT-PCR. Gene functions were characterized using overexpression in A. thaliana and silencing in G. hirsutum. Protein interactions were verified by yeast two-hybrid and luciferase complementation imaging (LCI) assays.

Results

We identified 14, 14, seven, and seven NRT2 proteins in G. hirsutum, G. barbadense, G. raimondii, and G. arboreum. Most NRT2 proteins were predicted in the plasma membrane. The NRT2 genes were classified into four distinct groups through evolutionary relationships, with members of the same group similar in conserved motifs and gene structure. The promoter regions of NRT2 genes included many elements related to growth regulation, phytohormones, and abiotic stresses. Tissue expression pattern results revealed that most GhNRT2 genes were specifically expressed in roots. Under low nitrate conditions, GhNRT2 genes exhibited different expression levels, with GhNRT2.1e being the most up-regulated. Arabidopsis plants overexpressing GhNRT2.1e exhibited increased biomass, nitrogen and nitrate accumulation, nitrogen uptake and utilization efficiency, nitrogen-metabolizing enzyme activity, and amino acid content under low nitrate conditions. In addition, GhNRT2.1e-silenced plants exhibited suppressed nitrate uptake and accumulation, hampered plant growth, affected nitrogen metabolism processes, and reduced tolerance to low nitrate. The results showed that GhNRT2.1e could promote nitrate uptake and transport under low nitrate conditions, thus effectively increasing nitrogen use efficiency (NUE). We found that GhNRT2.1e interacts with GhNAR2.1 by yeast two-hybrid and LCI assays.

Discussion

Our research lays the foundation to increase NUE and cultivate new cotton varieties with efficient nitrogen use.

Introduction

As an essential nutrient element, nitrogen (N) participates in the composition of nucleic acids, amino acids, chlorophyll, and many secondary metabolites (Akbudak, Filiz & Çetin, 2022). In agricultural production, nitrogen fertilizers are often over-applied to increase yields. However, less nitrogen fertilizer is absorbed by plants, and the rest is retained in soil or leaches into groundwater, causing increase in production costs, waste resources, and environmental problems (Zhou et al., 2022). Therefore, preventing the over-application of nitrogen fertilizer and increasing nitrogen use efficiency (NUE) is significant in protecting the environment and enhancing yield.

There are three forms of nitrogen available to plants including nitrate (NO3−), ammonium (NH4+), and organic nitrogen, among which NO3− is the most important form absorbed and utilized by plants (Ren, Zhou & Zhou, 2020). Nitrate absorption and transportation are primarily based on nitrate transporters (NRTs) (You et al., 2022). To accommodate the different nitrate concentrations in soil, plants have evolved low-affinity transport system (LATS) and high-affinity transport system (HATS) (Li et al., 2007; Léran et al., 2014). LATS mainly consists of nitrate transporter 1/peptide transporter (NRT1/PTR), while nitrate transporter 2 (NRT2) is part of HATS (Dechorgnat et al., 2011).

NRT2 genes are essential for improving NUE through efficient uptake and transport of nitrate. NRT2 genes have been investigated in A. thaliana (Orsel, Krapp & Daniel-Vedele, 2002), rice (Cai et al., 2008), barley (Trueman, Richardson & Forde, 1996), soybean (Amarasinghe et al., 1998), and maize (Lupini et al., 2016). In Arabidopsis, seven NRT2 members (AtNRT2.1-AtNRT2.7) were identified. AtNRT2.1 is primarily expressed in roots and significantly affects nitrate uptake (Wirth et al., 2007). Interestingly, elevated expression of AtNRT2.2 compensated for partial loss of nitrate uptake when AtNRT2.1 lost its ability to absorb nitrate (Li et al., 2007). AtNRT2.4 is primarily concerned with nitrate absorption in roots and transportation in the phloem (Kiba et al., 2012). AtNRT2.5 is expressed in leaves and roots, and its expression levels increase with the increase in nitrogen starvation (Lezhneva et al., 2014). AtNRT2.7 regulates nitrate storage in seeds (Chopin et al., 2007). Most NRT2 proteins depend on nitrate assimilation related (NAR2) proteins, and in Arabidopsis, most AtNRT2 proteins require binding to AtNAR2.1, whereas AtNRT2.7 can be transported alone (Kotur et al., 2012). A similar situation was demonstrated in rice, where OsNRT2.1, OsNRT2.2, and OsNRT2.3a are required to interact with OsNAR2.1, whereas OsNRT2.3b and OsNRT2.4 can absorb and transport nitrate alone (Wei et al., 2018).

As a cash crop, the growth and yield of cotton are regulated by nitrogen. Therefore, we can effectively improve cotton’s nitrogen uptake and utilization efficiency by mining nitrogen-efficient genes through molecular techniques (Magwanga et al., 2019). These essential genes will help us to maintain cotton yield while reducing nitrogen fertilizer input. In this research, we identified GhNRT2 genes and determined their evolutionary relationships, physicochemical properties, chromosomal location, gene duplication, collinearity relationship, conserved motifs, gene structure and cis-acting elements. After low nitrogen treatment, we observed the expression pattern of GhNRT2 genes and found that GhNRT2.1e was the most up-regulated gene. We characterized the GhNRT2.1e functions by overexpression in A. thaliana and silencing in G. hirsutum. Furthermore, we validated the interaction between GhNRT2.1e and GhNAR2.1 by yeast two-hybrid and LCI assays. This study lays a foundation for increasing NUE and breeding new cotton cultivars with high nitrogen using efficiency.

Materials and Methods

Plant materials and treatments

Cotton material used in the experiment was G. hirsutum L. TM-1. The seeds were germinated on moist filter paper until two cotyledons appeared. The seedlings continued to grow in the hydroponic conditions. Growing conditions were 16 h light at 28 °C and 8 h dark at 25 °C with 60% relative humidity. We changed the Hoagland nutrient solution every 5 days. We switched to the nitrogen-free nutrient solution at the three-leaf stage and continued for 1 week. The NO3− concentration in Hoagland’s nutrient solution was then adjusted to 0.25 mM for low nitrogen treatment. Root tissues were collected at 0, 1, 3, 6, 12, 24, and 48 h after treatment and used for RNA extraction. The Arabidopsis ecotypes Columbia-0 and Nicotiana benthamiana were grown in nutrient soil and vermiculite under light conditions for 16 h at 22 °C and 8 h of dark conditions at 18 °C with 60% relative humidity.

Identification of NRT2 genes in cotton

To research NRT2 family members in cotton, CottonGEN (http://www.cottongen.org/) was used to download genome files of G. barbadense (ZJU_v1.1), G. hirsutum (ZJU_v2.1), G. raimondii (JGI_v1.0), and G. arboreum (CRI_v1.0) (Yu et al., 2014). The protein sequences of AtNRT2 in Arabidopsis were downloaded from TAIR (https://www.arabidopsis.org/) (Berardini et al., 2015). We used AtNRT2 proteins as query sequences to identify NRT2 proteins in four cotton species by BlastP (Chen et al., 2020). All identified NRT2 proteins were subjected to domain analysis by Pfam (http://pfam.xfam.org/) (El-Gebali et al., 2019). We referred to the nomenclature of the rapeseed NRT2 genes for naming the cotton NRT2 genes (Tong et al., 2020). The physicochemical properties of NRT2 proteins including amino acid length, molecular weight (MW) and isoelectric point (pI) were analyzed by ExPASy-ProtParam (http://web.expasy.org/protparam/) (Wilkins et al., 1999). The subcellular localization of NRT2 proteins was predicted using WOLF-PSORT (https://wolfpsort.hgc.jp/) (Horton et al., 2007). The transmembrane helices of GhNRT2 proteins were predicted by TMHMM 2.0 online server (https://services.healthtech.dtu.dk/service.php?TMHMM-2.0/) (Krogh et al., 2001).

Phylogenetic analysis, chromosomal location and synteny analysis of NRT2 genes

The evolutionary relationships of NRT2 proteins were indicated by phylogenetic analysis. Multiple sequence alignment of NRT2 proteins in A. thaliana, G. hirsutum, G. raimondii, G. arboreum, and G. barbadense was performed through ClustalW (Larkin et al., 2007). The phylogenetic tree was constructed using the Neighbor-Joining (NJ) method in MEGA 7.0 software and drawn by iTOL (https://itol.embl.de/) (Kumar, Stecher & Tamura, 2016; Letunic & Bork, 2021). Chromosome location of NRT2 genes was acquired from the gff3 file downloaded by CottonGen (https://www.cottongen.org/), and TBtools was used to visualize the distribution of NRT2 genes on chromosomes (Chen et al., 2020). The synteny relationship of NRT2 genes was investigated using MCScanX (Wang et al., 2012). To investigate the selection pressure between homologous gene pairs, we used TBtools to calculate nonsynonymous substitution (Ka) and synonymous substitution (Ks) rates for duplicated gene pairs (Chen et al., 2020).

Gene structure, conserved motif and cis-acting elements analysis

The structure of GhNRT2 genes were investigated by GSDS (http://gsds.gao-lab.org/) (Hu et al., 2015). The conserved motifs of GhNRT2 proteins were predicted by MEME (http://meme-suite.org/tools/meme) (Bailey et al., 2015). The 2,000 bp upstream sequences of the initiation codons of GhNRT2 genes were extracted, and predicted cis-acting elements using the PlantCARE database (http://bioinformatics.psb.ugent.be/webtools/plantcare/html/) (Lescot et al., 2002).

Analysis of expression patterns of GhNRT2 genes

To analyze the expression patterns of GhNRT2 genes in different tissues, we downloaded RNA-seq data (accession number: PRJNA490626) of roots, stems, leaves, petals, receptacles, sepals, bracts, ovules, and fibers of G. hirsutum TM-1 from the Cotton Omics Database (http://cotton.zju.edu.cn) (Hu et al., 2019). The expression levels of GhNRT2 genes were visualized by TBtools based on log2 (FPKM+1) values (Chen et al., 2020). To validate the results of RNA-seq, we collected roots, stems, and leaves tissues from TM-1 plants at the three-leaf stage for RNA extraction.

RNA extraction and qRT-PCR analysis

We obtained total RNA by RNAprep Pure Plant kit (Tiangen, Beijing, China) and measured the quantity and quality of RNA samples by spectrophotometer. The RNA was reverse transcribed into cDNA using StarScript II First-strand cDNA Synthesis kit (GenStar, Beijing, China). The quality of the cDNA was examined by PCR using the GhActin gene as an internal gene, and better quality cDNA was used for qRT-PCR. The qRT-PCR was performed using fluorescence quantitative kit 2×RealStar Green Fast Mixture with ROX II (GenStar, Beijing, China), and the GhActin gene was used as an internal gene. The experiment was performed with three replicates. We calculated the relative expression levels of genes according to the 2−ΔΔCT method (Schmittgen & Livak, 2008). For the accuracy of the results, we also selected the GhHis3 gene as the internal gene for verification. The results using GhHis3 as an internal control were similar to those of GhActin, indicating the accuracy of the results and the availability of both internal reference genes for qRT-PCR analysis. Specific primers for all genes were designed by NCBI Primer-blast (http://www.ncbi.nlm.nih.gov/tools/primer-blast/) (Table S1).

Overexpression vector construction and transformation

To characterize the function of GhNRT2.1e, we amplified the GhNRT2.1e gene using primers PRI101-GhNRT2.1e-F (SalI) and PRI101-GhNRT2.1e-R (BamHI) (Table S1). The PCR products were inserted into the pRI101 vector containing the Cauliflower mosaic virus 35S promoter forming an overexpression recombinant. The 35S: GhNRT2.1e overexpression recombinant was transformed to wild-type A. thaliana (Colombia-0) through Agrobacterium tumefaciens flower soaking method. A total of 50 mg/L kanamycin was added to the MS medium for screening transformed T1 and T2 generations (Clough & Bent, 1998). We extracted RNA and genomic DNA from transgenic A. thaliana and used AtActin as an internal gene to test the quality of cDNA and genomic DNA. We performed PCR analysis of genomic DNA from transgenic A. thaliana using primers 35S: PRI101-F and PRI101-GhNRT2.1e-R (BamHI). After PCR and qRT-PCR analysis in the T2 generation, the highly expressed T3 generation was used for phenotypic research. A. thaliana grown in vermiculite were provided with 0.25 and 2.5 mM of nitrate, creating a nitrogen deficiency and normal supply. Plants were grown for 6 weeks by irrigating with a nutrient solution once a week.

Virus-induced gene silencing

We performed virus-induced gene silencing (VIGS) using tobacco rattle virus (TRV) to further investigate the function of GhNRT2.1e (Gao et al., 2011). A sequence fragment of the GhNRT2.1e gene was amplified using specific primers VIGS-GhNRT2.1e-F (EcoRI) and VIGS-GhNRT2.1e-R (BamHI) and inserted into the TRV2 vector to form TRV2:GhNRT2.1e (Table S1). Negative controls were wild-type and TRV2:00, while the effectiveness of the vector was tested by phytoene desaturase (PDS). Cotton seedlings were used for injection when both cotyledons were fully expanded. The NO3− concentration was adjusted to 0.25 and 2.5 mM to provide plants with nitrogen deficiency and normal supply. Two weeks after plant growth, RNA from TRV2:GhNRT2.1e, TRV2:00, and WT was extracted, and the silencing efficiency of GhNRT2.1e was determined by qRT-PCR.

Plant physiological and biochemical evaluation

We performed physiological and biochemical evaluations of Arabidopsis and VIGS plants. Plant samples were dried, and the dry weight was measured using a balance. The relative chlorophyll contents were measured with the SPAD analyzer. The Kjeldahl method measured the total nitrogen contents (Singh et al., 2020). The salicylic acid method measured the nitrate content (Zhao & Wang, 2017). Iqbal et al. (2020a) method calculated total nitrogen accumulation, nitrogen utilization efficiency (NUtE), and nitrogen uptake efficiency (NUpE). Nitrate reductase (NR), glutamine synthetase (GS), and glutamate synthase (GOGAT) activities, as well as amino acid content were measured by assay kits (Solarbio, Beijing, China). In addition, we measured the activity of antioxidants and oxidants: superoxide dismutase (SOD), peroxidase (POD), catalase (CAT), and malondialdehyde (MDA) by detection kits (Solarbio, Beijing, China).

Yeast two-hybrid assay

In Arabidopsis, the interaction between AtNRT2.1 and AtNAR2.1 has been verified. We used the protein sequence of AtNAR2.1 to verify whether this interaction exists in G. hirsutum. Finally, we selected the homolog of AtNAR2.1 and named it as GhNAR2.1. The interaction between GhNRT2.1e and GhNAR2.1 was verified by yeast two-hybrid assays. The specific primers pPR3-N-GhNRT2.1e-F (BamHI)/pPR3-N-GhNRT2.1e-R (EcoRI) and pBT3-C-GhNAR2.1-F (XbaI)/pBT3-C-GhNAR2.1-R (NcoI) were used to amplify the GhNRT2.1e and GhNAR2.1 coding sequences and inserted them into the pPR3-N and pBT3-C vectors to form the pPR3-N-GhNRT2.1e and pBT3-C-GhNAR2.1 constructs (Table S1). The two constructs were transferred into yeast NMY51, the transformants formed were grown on an SD-LW medium, and positive transformants were selected for serial dilution (100, 10−1, 10−2, 10−3) (Liu et al., 2014). The different yeast dilutions were cultured on SD-LW and SD-AHLW media at 30 °C.

Luciferase complementation imaging assay

To verify whether GhNRT2.1e and GhNAR2.1 interact in vivo, we tested by LCI assay. The coding sequences of GhNRT2.1e and GhNAR2.1 were amplified using the specific primers nLUC-GhNRT2.1e-F (BamHI)/nLUC-GhNRT2.1e-R (SalI) and cLUC-GhNAR2.1-F (BamHI)/cLUC-GhNAR2.1-R (SalI), and they were inserted into the nLUC and cLUC vectors to form the nLUC-GhNRT2.1e and cLUC-GhNAR2.1 constructs. The Agrobacterium inoculum containing the constructs was mixed in equal amounts and injected into the tobacco (Xie et al., 2020). Tobacco was grown under dark conditions for 24 h, followed by 2 days in normal environments. The undersides of the leaves were coated with 1 mM Luciferin sodium salt, and after 10 min of dark treatment, fluorescent pictures were obtained with a CCD imaging device (Li et al., 2021).

Results

Identification of NRT2 genes in cotton

We identified 14, 14, seven, and seven proteins in G. hirsutum, G. barbadense, G. raimondii, and G. arboreum using seven AtNRT2 proteins as query sequences, which were named according their homologs in A. thaliana. We further analyzed the physicochemical properties of NRT2 proteins (Table S2). In G. hirsutum, the protein lengths ranged from 341aa (GhNRT2.1f) to 542aa (GhNRT2.4a and GhNRT2.4b). The isoelectric point ranged from 8.04 (GhNRT2.7a) to 9.51 (GhNRT2.1b), indicating that all GhNRT2 proteins are basic proteins. The subcellular localization of most GhNRT2 proteins was predicted at the plasma membrane, while GhNRT2.7a and GhNRT2.7b were localized in the vacuole. The difference in subcellular localization results suggested that GhNRT2 proteins may perform transport functions at different locations. The GhNRT2 proteins have seven to 12 transmembrane helices, further supporting the predicted subcellular localization results.

Phylogenetic analysis of NRT2 proteins in cotton

The phylogenetic tree of NRT2 proteins was constructed by the NJ method in MEGA7.0 software (Fig. 1). The NRT2 genes were divided into four groups, named Group 1 to Group 4. Group 1 consisted of GhNRT2.1s, GhNRT2.4s, and their homologous genes. Group 2 was composed of GhNRT2.3s and their homologous genes. GhNRT2.7s and their homologs were located in group 3. GhNRT2.5s and their homologs were found in group 4. AtNRT2.2 and AtNRT2.6 were located in groups 1 and 2, respectively, but their homologous genes were not identified in cotton. The NRT2 proteins of the At subgenome of allotetraploid cotton share high homology with those from G. arboreum. The NRT2 proteins of the Dt subgenome share a high degree of homology with those of G. raimondii. The results further verified that the allotetraploid cotton was produced by recombining two diploid species of cotton.

Figure 1 Phylogenetic analysis of NRT2 proteins.

The phylogenetic tree was constructed using the neighbor-joining (NJ) method in MEGA 7.0 software. Different groups were marked with different colors. AT, A. thaliana; Gr, G. raimondii; Ga, G. arboreum; Gh, G. hirsutum and Gb, G. barbadense.

Chromosomal location of GhNRT2 genes

The distribution of GhNRT2 genes on chromosomes was mapped through gene location information (Fig. 2). In G. hirsutum, fourteen GhNRT2 genes were distributed on twelve chromosomes. Such as chromosomes A05, A07, A08, A09, A12, D05, D07, D08, D09, and D12 each contained one GhNRT2 gene, while chromosomes A03 and D03 each contained two genes.

Figure 2 Chromosomal location of GhNRT2 genes in G. hirsutum.

The chromosomal locations of the GhNRT2 genes were mapped by TBtools software. The black lines indicate the location of the genes on the chromosomes. The scale bar indicates the chromosome length in the base pair (bp).

Gene duplication and synteny analysis of NRT2 gene in cotton

We performed gene duplication and syntenic relationship analysis in NRT2 genes and identified two tandem repeats in G. hirsutum, GhNRT2.1a/GhNRT2.3a and GhNRT2.1b/GhNRT2.3b. In addition, we identified 22 segmental duplications and 119 WGD for the GhNRT2 genes (Table S3). Synteny analysis revealed a high collinearity relationship between NRT2 genes. We identified 22, 56, 30, 39, 36, 29, and 27 paralogous or orthologous gene pairs from seven combinations (Gh-Gh, Gh-Gb, Gb-Gb, Gb-Gr, Gh-Gr, Gb-Ga, and Gh-Ga), respectively (Fig. 3). In addition, only GhNRT2.7a/GbNRT2.7a had a Ka/Ks ratio >1, indicating that positive selection was experienced. In contrast, the Ka/Ks of other gene pairs were <1, demonstrating that almost all NRT2 genes underwent purifying selection (Table S4).

Figure 3 Syntenic relationship of NRT2 genes in G.harsutum, G.barbadense, G.arboreum, and G.raimondii.

The syntenic relationships of the NRT2 genes were mapped by the MCScanX program in TBtools software. The lines represented by various colors indicate the syntenic regions around the NRT2 genes. Gr, G. raimondii; Ga, G. arboreum; Gh, G. hirsutum and Gb, G. barbadense.

Gene structure and conserved motif of GhNRT2 genes

To further study the structural diversity of GhNRT2 genes, we analyzed their exon-intron structure. Structural analysis of GhNRT2 genes revealed that GhNRT2.1s and GhNRT2.4s contained three exons and two introns, while GhNRT2.3s, GhNRT2.5s, and GhNRT2.7s all contained only two exons and one intron (Fig. 4C). Furthermore, we found that genes in the same group have similar structures, probably because the GhNRT2 genes are highly conserved in structure. The analysis of the gene structure also provided additional evidence for the evolutionary relationship of NRT2 genes. We identified conserved motifs in GhNRT2 proteins by MEME. We found that almost all GhNRT2 proteins have motifs 1, 2, 3, 4, 5, 6, 7, 8, and 10, suggesting that these proteins are highly conserved and have similar functions. GhNRT2.7a and GhNRT2.7b did not contain motif 9, and GhNRT2.1f was missing in motifs 7, 8, 9, and 10, so we speculated that these proteins might have specific functions (Fig. 4B). To verify the sequence characteristics of GhNRT2 proteins, we performed multiple sequence alignments of GhNRT2 protein sequences by DNAMAN (Fig. 4D). All GhNRT2 proteins contained MFS and NNP motifs, demonstrating that they belong to the MFS and NNP families.

Figure 4 Phylogenetic relationship, conserved motifs, gene structure, and multiple sequence alignment of GhNRT2 proteins.

(A) Phylogenetic analysis of GhNRT2 proteins. (B) Conserved motifs of GhNRT2 proteins. The MEME software identified the conserved motifs. Different colored boxes represented different motifs. (C) Exon-intron structures of the GhNRT2 genes. Gene structure was analyzed by GSDS software. (D) Multiple sequence alignment of GhNRT2 proteins. Amino acid sequence alignment was performed by DNAMAN software. MFS motifs and NNP motifs were boxed in red.

Cis-acting element analysis of GhNRT2 genes

The promoter cis-acting elements of GhNRT2 genes were examined by PlantCARE software to elucidate potential regulatory mechanisms (Fig. 5). We classified the examined cis-acting elements into three categories based on their biological functions (Table S5). The first category was phytohormone-related elements, of which MeJA, GA, auxin, SA, and ABA contained 30, 18, 10, 9, and 27, respectively. The second category was stress response elements. We found the highest number of anaerobic induction elements with 34. There was only one anoxic specific inducibility element, and it was present in the promoter region of GhNRT2.3b. The third category was growth-related elements. The number of meristem expression elements was the most abundant, containing seven. Based on the results, it is hypothesized that GhNRT2 genes may participate in plant growth, phytohormone regulation, and abiotic stress.

Figure 5 Predicted cis-acting elements in the promoters of GhNRT2 genes.

The cis-acting elements were predicted from the PlantCARE database. Different colored boxes represented different cis-acting elements.

Tissue-specific expression pattern analysis of GhNRT2 genes

The tissue expression pattern of GhNRT2 genes were revealed using RNA-seq data (accession number: PRJNA490626) (Fig. 6). We found that the expression levels of GhNRT2.1f, GhNRT2.3a, and GhNRT2.4a were not detected in all tissues (FPKM < 1). The expression levels of GhNRT2.1a-e were extremely high in roots. GhNRT2.3b and GhNRT2.4b were expressed only in the roots, but their expression levels were low. GhNRT2.5a and GhNRT2.5b had high expression levels in several tissues, with GhNRT2.5a primarily expressed in epicalyx, root, and leaf, and GhNRT2.5b had high levels in petal, pistil, and sepal. The expression levels of GhNRT2.7a and GhNRT2.7b were higher in leaves and ovules. To validate the RNA-seq results, we analyzed the expression patterns of GhNRT2 genes in root, stem, and leaf by qRT-PCR (Fig. 7). We designed specific primers for each GhNRT2 gene to detect accurate gene expression levels. We found similar results of qRT-PCR and RNA-seq analysis, further validating the expression profile of GhNRT2 genes. The GhNRT2 genes were highly expressed in roots, suggesting that they may function in nitrate uptake. The qRT-PCR assay showed that GhNRT2.5a and GhNRT2.5b had the highest expression in roots. Combining the results of RNA-seq and qRT-PCR, we found that GhNRT2.5a and GhNRT2.5b may be expressed in roots and aerial parts, and their expression patterns were similar to those of AtNRT2.5 (Lezhneva et al., 2014). GhNRT2.7a and GhNRT2.7b showed high expression in leaves and ovules, and they may have potential functions in nitrate storage.

Figure 6 RNA-seq data heat map of GhNRT2 genes expression in eleven different tissues.

The differences in expression of the GhNRT2 genes are shown in different colors. DPA, days post anthesis.

Figure 7 The relative expression levels of GhNRT2 genes in root, stem, and leaf were determined by qRT-PCR.

GhNRT2.1a expression in roots was set to 1 and used as a control to reflect the relative expression of the GhNRT2 genes in roots, stems, and leaves. The GhActin gene was used as the internal reference gene.

Expression patterns of GhNRT2 genes at low nitrate concentrations

After 1 week of nitrogen-free treatment, cotton seedlings were resupplied with 0.25 mM NO3− to investigate the expression pattern of GhNRT2 genes at low nitrate concentrations. We found that GhNRT2.7a and GhNRT2.7b were barely expressed in the roots by tissue-specific analysis. Therefore, we selected nine GhNRT2 genes with high or specific expression in the roots and analyzed their expression levels at low nitrate concentrations by qRT-PCR (Fig. 8). The GhNRT2 genes exhibited different expression levels after NO3− resupply. Interestingly, only the GhNRT2.1(a-e) genes were significantly up-regulated and peaked at 3 or 6 h before gradually decreasing. However, the expression levels of GhNRT2.4b, GhNRT2.5a and GhNRT2.5b genes were reduced after NO3− resupply, and the reduction was more pronounced for GhNRT2.5a and GhNRT2.5b. GhNRT2.3b may not be sensitive to nitrate resupply and is weakly upregulated. It may be because the GhNRT2.1(a-e) proteins belong to the inducible high-affinity transport system (iHATS) which is induced by nitrate supply. In contrast, GhNRT2.4b, GhNRT2.5a and GhNRT2.5b proteins belong to constitutive high-affinity transport systems (cHATS), which are active in plants not supplied with nitrate (Lezhneva et al., 2014). The GhNRT2.1e showed the most significant up-regulation of all genes, and its expression level reached the highest at 6 h, which was about 180-fold increase compared to 0 h. GhNRT2.1e may be a critical gene to promote the absorption and transport of nitrate in cotton.

Figure 8 The relative expression levels of GhNRT2 genes were analyzed by qRT-PCR under low nitrate conditions.

Cotton seedlings were grown in a nitrogen-free nutrient solution for 1 week and then resupplied with 0.25 mM NO3−. Root tissues were collected at 0, 1, 3, 6, 12, 24, and 48 h after treatment, respectively. The GhActin gene was used as the internal control.

Phenotypic, physiological, and biochemical evaluation of GhNRT2.1e-overexpressed Arabidopsis plants

To elucidate the function of GhNRT2.1e, we cloned the coding sequence of GhNRT2.1e into the pRI101 vector containing the Cauliflower mosaic virus 35S promoter, and overexpressed it in Arabidopsis. We extracted RNA and genomic DNA from T2 generation of overexpressed lines and analyzed the expression level of GhNRT2.1e under a normal nitrate environment by qRT-PCR and PCR (Figs. 9B and S1). Results indicated that GhNRT2.1e had been successfully transferred into Arabidopsis and could be stably expressed. Three T3 generation overexpressed lines were screened for further functional studies of GhNRT2.1e. Plants were supplied with 0.25 mM and 2.5 mM of nitrate to create a nitrogen deficiency and normal supply. After 6 weeks of growth, plants were evaluated for phenotypic, physiological, and biochemical traits. There were no phenotypic differences between overexpressed lines and WT under 2.5 mM nitrate conditions. However, the overexpressed lines exhibited better growth than the WT when watered with 0.25 mM NO3− (Fig. 9A). We further measured dry weight to reflect differences between plants and found that the overexpressed lines had higher biomass than WT (Fig. 9C). To reflect physiological differences between plants, we measured nitrogen and nitrate content (Figs. 9D and 9E). There were no differences between plants under normal conditions, but overexpressed lines had higher levels of nitrogen and nitrate at low nitrate concentrations. Additionally, we found that nitrogen accumulation in overexpressed lines was increased under low nitrate conditions (Fig. 10A). We also calculated the NUpE and NUtE of plants and found that the NUpE and NUtE were higher in the overexpressed lines (Figs. 10B and 10C). We speculated that the overexpressed lines may promote nitrate uptake, allowing plants to accumulate and utilize more nitrogen and nitrate. To further analyze nitrate uptake and utilization in plants, we measured nitrogen-metabolizing enzyme activity (Figs. 10D–10F). Nitrogen-metabolizing enzyme activity was higher in the overexpressed lines at low nitrate concentrations, but there were no apparent differences at normal concentrations. We also observed higher amino acid contents in overexpressed lines (Fig. 10G). In conclusion, Arabidopsis undergoes low-affinity transport under normal conditions, and GhNRT2.1e plays a minor role, so there is no apparent difference between overexpressed Arabidopsis and WT. In contrast, under low nitrate conditions, GhNRT2.1e plays an essential role as a high-affinity transporter protein to absorb more nitrate into nitrogen metabolism, which produces more nitrogen assimilation products for plant growth with the action of nitrogen metabolizing enzymes.

Figure 9 Phenotypic observation and identification of GhNRT2.1e-overexpressed Arabidopsis.

(A) Phenotypes of wild-type (WT) and overexpression lines grown under 0.25 mM nitrate conditions. Each small pot contained three plants. (B) The relative expression levels of GhNRT2.1e in wild-type (WT) and three T2 generations overexpressed lines were analyzed by qRT-PCR under normal nitrate conditions. The AtActin gene was used as the internal control. Quantitative determination of biomass (C), total nitrogen content (D), nitrate content (E), and total nitrogen accumulation (F) of WT and overexpression lines. The values are means ± standard deviation (SD) of ten replicates. Student’s t-test was used to analyze the significance of differences. **P < 0.01 indicates significant differences between WT and overexpression lines. “CK” represents a nutrient solution with a nitrate concentration of 2.5 mM; “Low Nitrate” represents a nutrient solution with a nitrate concentration of 0.25 mM; “WT” represents the wild type; “OE” represents overexpression line.

Figure 10 Physiological and biochemical analysis of GhNRT2.1e-overexpressed Arabidopsis.

Nitrogen uptake efficiency (NUpE) (A), nitrogen utilization efficiency (NUtE) (B), Nitrate reductase (NR) activity (C), Glutamine synthetase (GS) activity (D), Glutamate synthetase (GOGAT) activity (E), and amino acid content (F) of WT and overexpression lines grown under 2.5 and 0.25 mM NO3− conditions. The values are means ± standard deviation (SD) of ten replicates. Student’s t-test was used to analyze the significance of differences. **P < 0.01 indicates significant differences between WT and overexpression lines. “CK” represents a nutrient solution with a nitrate concentration of 2.5 mM; “Low Nitrate” represents a nutrient solution with a nitrate concentration of 0.25 mM; “WT” represents the wild type; “OE” represents overexpression line.

Evaluation of morphology, physiological indexes, and biochemical characters of GhNRT2.1e-silenced plants

To further elucidate the function of GhNRT2.1e, we selected the TRV vector for the VIGS assay. To verify the effectiveness of the vector, we injected TRV2:PDS into cotton leaves. Cotton leaves showed an albino phenotype after 14 days TRV2:PDS injection, indicating that the vector used was effective (Fig. 11A). After injection, plants were given nutrient solutions with a nitrate concentration of 2.5 and 0.25 mM to create normal and low nitrate environments, respectively. To evaluate gene silencing efficiency, we determined the expression levels of GhNRT2.1e in WT, TRV2:00, and TRV2:GhNRT2.1e plants by qRT-PCR after 2 weeks of plant growth (Fig. 11C). The expression levels of the GhNRT2.1e gene were not significantly different in WT and TRV2:00, indicating that the injection of an empty vector did not affect the plants. However, the expression level of the GhNRT2.1e gene was significantly down-regulated in TRV2:GhNRT2.1e plants compared with WT and TRV2:00, which indicated that gene silencing was successful. After 4 weeks of growth, there were no apparent differences between plants under 2.5 mM nitrate conditions. However, silenced plants showed dwarf and slow growth under low nitrate conditions compared to WT and TRV2:00 (Fig. 11B). To further analyze the morphological differences between VIGS and control plants, we evaluated physiological traits including dry weight, nitrogen content, and nitrate content (Figs. 11D–11F). In the low nitrate environment, VIGS plants had lower dry weight, nitrogen, and nitrate content than WT and TRV2:00. We further analyzed NUpE and NUtE and found that the uptake and utilization efficiencies of VIGS plants were low as compared to WT and TRV2:00 (Figs. 12A and 12B). Additionally, we also found lower total nitrogen accumulation in VIGS plants (Fig. 12C). The differences exhibited by the plants in normal and low nitrate environments could be due to the high nitrate concentration in the normal environment, where plants mainly undergo low-affinity transport. However, GhNRT2.1e is a high-affinity transporter protein that may not function or have less effect in environments with high nitrate concentrations. In low nitrate environments, plants mainly undergo high-affinity transport, and GhNRT2.1e plays an essential role as a critical gene. Therefore, nitrate uptake by GhNRT2.1e silenced plants is inhibited, which affects the accumulation of nitrogen and nitrate and ultimately reduces plant biomass. We also found that chlorophyll content decreased in VIGS plants under low nitrate conditions (Fig. 12D). Nitrogen is a vital chlorophyll component, essential for photosynthesis and plant growth. Therefore, reducing chlorophyll content also reflects the inhibition of nitrogen uptake and utilization by VIGS plants. We measured nitrogen metabolizing enzyme activity to reflect whether nitrate uptake by silenced plants was affected (Figs. 12E–12G). The activity of nitrogen metabolizing enzymes was reduced in VIGS plants compared with controls. We also observed that the amino acid content was reduced in VIGS plants (Fig. 12H). These results suggested that silencing of GhNRT2.1e affects nitrate uptake by plants, thereby reducing nitrogen-metabolizing enzyme activity and amino acid content. We assessed the concentrations of oxidants and antioxidant enzymes in plants including CAT, POD, SOD, and MDA (Figs. 12I–12L). Under low nitrate conditions, the concentrations of CAT, POD, and SOD in VIGS plants decreased, and the concentration of MDA increased compared with control plants. Under stress conditions, elevated levels of ROS lead to membrane damage and an increase in MDA levels. High concentrations of MDA in silenced plants indicated increased peroxidation of plant cells and oxidative damage to the plants, as further showed by reduced antioxidant enzyme activity. These results showed that plants suffer from nutrient stress in low nitrate environments, and GhNRT2.1e perform high-affinity transport to maintain a partial nitrate supply. However, silencing of GhNRT2.1e resulted inhibition of nitrate uptake and reduced tolerance to low-nitrate environments, leading to oxidative damage in plants.

Figure 11 Phenotypic observation and identification of silenced plants.

(A) Albino phenotype appearance on the leaves of the TRV2:PDS infused plants. (B) The phenotype of WT, TRV2:00, and TRV2:GhNRT2.1e plants grown at 0.25 mM nitrate concentration. (C) The expression levels of GhNRT2.1e in WT, TRV2:00, and TRV2:GhNRT2.1e plants grown at 0.25 mM and 2.5 mM nitrate concentrations were analyzed by qRT-PCR. The GhActin gene was used as the internal control. Comparison of dry weight (D), total nitrogen content (E), and nitrate content (F) of WT, TRV2:00, and TRV2:GhNRT2.1e plants. The values are means ± standard deviation (SD) of ten replicates. Student’s t-test was used to analyze the significance of differences. **P < 0.01 indicates significant differences between GhNRT2.1e-silenced plants and control plants. “WT” represents the wild type; “TRV2:00” represents the plants carrying control the TRV2 empty vector; “TRV2:GhNRT2.1e” represents the GhNRT2.1e-silenced plants; “CK” represents a nutrient solution with a nitrate concentration of 2.5 mM; “Low Nitrate” represents a nutrient solution with a nitrate concentration of 0.25 mM.

Figure 12 Physiological and biochemical evaluation of GhNRT2.1e-VIGS cotton plants.

Quantitative determination of nitrogen uptake efficiency (NUpE) (A), nitrogen utilization efficiency (NUtE) (B), total nitrogen accumulation (C), chlorophyll content (D), Nitrate reductase (NR) activity (E), Glutamine synthetase (GS) activity (F), Glutamate synthetase (GOGAT) activity (G), amino acid content (H), catalase (CAT) activity (I), peroxidase (POD) activity (J), superoxide dismutase (SOD) activity (K) and malondialdehyde (MDA) concentration (L) in GhNRT2.1e-silenced and control plants grown under 2.5 mM NO3− and 0.25 mM NO3− conditions. The values are means ± standard deviation (SD) of ten replicates. Student’s t-test was used to analyze the significance of differences. **P < 0.01 indicates significant differences between GhNRT2.1e-silenced plants and control plants. “WT” represents the wild type; “TRV2:00” represents the plants carrying control the TRV2 empty vector; “TRV2:GhNRT2.1e” represents the GhNRT2.1e-silenced plants; “CK” represents a nutrient solution with a nitrate concentration of 2.5 mM; “Low Nitrate” represents a nutrient solution with a nitrate concentration of 0.25 mM.

GhNRT2.1e interacts with GhNAR2.1

We performed a yeast two-hybrid assay to verify interaction between GhNRT2.1e and GhNAR2.1 in cotton as we identified the homologous gene GhNAR2.1 of AtNAR2.1 in G.hirsutum. On SD-LW medium, yeast carrying pPR3-N-GhNRT2.1e/pBT3-C-GhNAR2.1, pPR3-N-GhNRT2.1e/pBT3-C, and pPR3-N/pBT3-C-GhNAR2.1 could grow normally (Fig. 13A). On the SD-AHLW medium, only yeast carrying pPR3-N-GhNRT2.1e/pBT3-C-GhNAR2.1 could grow normally (Fig. 13B). Among them, pPR3-N and pBT3-C were empty vectors, and the constructs pBT3-C-GhNAR2.1 and pPR3-N-GhNRT2.1e were connected with empty vectors as negative controls. The results indicated that GhNRT2.1e interacts with GhNAR2.1, which was further verified by LCI assay. SPL3-nLUC and FHY3-cLUC served as positive controls (Fig. 13C). In tobacco leaves, only GhNRT2.1e-nLUC/GhNAR2.1-cLUC could fluoresce (Fig. 13D). The results showed that GhNRT2.1e and GhNAR2.1 could also interact in vivo.

Figure 13 The interaction between GhNRT2.1e and GhNAR2.1 was verified by yeast two-hybrid assay and luciferase complementation imaging assays.

(A) Yeast cells were grown on SD-LW (SD-Leu-Trp) medium. (B) Yeast cells were grown on SD-AHLW (SD-Ade-His-Leu-Trp) medium. (C) Imaging of luciferase complementation between SPL3-nLUC and FHY3-cLUC as a positive control. (D) Imaging of luciferase complementation between GhNRT2.1e and GhNAR2.1.

Discussion

Identification and characterization of the NRT2 genes

Nitrogen fertilizer can promote cotton growth but over-application of nitrogen fertilizer causes waste of resources and environmental pollution. Therefore, it is necessary to apply nitrogen fertilizer rationally and improve the NUE to save resources and protect the environment (Xu, Fan & Miller, 2012). We can explore key genes regulating nitrogen uptake and utilization through molecular methods and breeding new cotton cultivars to improve NUE. NRT2 proteins have been found in plants, which can uptake and transport nitrate at low nitrate concentrations and are particularly important for increasing NUE (Chen et al., 2016).

In this research, the numbers NRT2 genes were twice in tetraploid cotton as in diploid cotton species, verifying that G. hirsutum was developed by crossing two diploid cotton species (Huang et al., 2020). Gene duplication promotes functional differentiation, improves adaptation to the environment during evolution, and expands the gene family (Cannon et al., 2004). In the replication analysis of NRT2 genes, segmental duplication, tandem repeats, and WGD were observed to expand the NRT2 gene family. Besides, the NRT2 genes may have diverged during the duplication process to generate new functions. Almost all NRT2 genes have undergone purifying selection during evolution. Similar results were observed for changes in the number of NRT2 proteins in Brassica families (Tong et al., 2020). Purification selection is essential to reduce deleterious mutations in the NRT2 genes and increase their stability. We performed subcellular localization predictions and found that most of the cotton NRT2 proteins were present in the plasma membrane, and only NRT2.7 was on the vacuolar membrane. These results are similar to the localization of Arabidopsis NRT2 proteins (Lezhneva et al., 2014). Furthermore, our prediction of the transmembrane helix of NRT2 proteins confirmed their localization to membranes. The plasma membrane controls the inflow and outflow of substances during the exchange of ions, metabolites, and nutrients (Barberon et al., 2014). Therefore, NRT2 proteins on membranes are essential for cellular uptake and transport of nitrate. Predicting cis-acting elements helps us speculate on gene function (Feng et al., 2021). Some hormone-related elements were found in the promoter region of GhNRT2 genes. Among those, ABA, auxin, and cytokinin are closely related to regulating nitrogen signaling and nitrogen uptake (Kiba et al., 2011). Nitrate controls ABA accumulation in root tips, while ABA regulates lateral root formation by nitrate (Ondzighi-Assoume, Chakraborty & Harris, 2016). Arabidopsis can regulate auxin levels in their roots according to their nitrogen status (Kiba et al., 2011). In watermelon, SA regulates nitrate uptake and assimilation (Vega, O’Brien & Gutiérrez, 2019). The expression of GhNRT2 genes in roots and their response to low nitrate may be related to ABA, auxin, and SA response elements and phytohormone regulation of nitrogen signaling.

Expression analysis of the GhNRT2 genes

Tissue differential expression analysis facilitates the investigation of gene functions (Yang et al., 2019). We identified differential expression of GhNRT2 genes by RNA-seq and qRT-PCR (Figs. 6 and 7). The GhNRT2 genes were expressed in roots, which might be related to nitrate uptake by the roots. Both GhNRT2.5a and GhNRT2.5b had the high expression in several tissues and may be implicated in root uptake and inter-tissue transport. GhNRT2.7a and GhNRT2.7b were expressed mainly in ovules and leaves and may participate in nitrate storage and accumulation. There was a similar expression in Arabidopsis, where AtNRT2.7 was expressed in leaves and seeds, while other genes were expressed in roots (Orsel, Krapp & Daniel-Vedele, 2002). Similar situations were identified in cassava and rapeseed (Du et al., 2022; You et al., 2022). The similarity in the expression of homologous genes in different species suggests that NRT2 genes are relatively conserved during evolution and have an essential role in plants. The tissue specificity of GhNRT2 genes suggested that they function at different sites to jointly promote nitrate uptake, transport, storage, and utilization.

The expression levels of GhNRT2 genes were investigated after nitrate supplementation. The GhNRT2.1(a-e) genes were strongly induced by nitrate, whereas GhNRT2.4b, GhNRT2.5a and GhNRT2.5b were significantly repressed (Fig. 8). The cHATS is active without nitrate supply and can initiate nitrate-induced genes (Kiba et al., 2012). In lack of nitrate supply, iHATS is barely expressed, whereas expression is high hours of nitrate induction (Kiba et al., 2012; Orsel, Krapp & Daniel-Vedele, 2002). In A. thaliana, iHATS activity is mainly dependent on AtNRT2.1 (Li et al., 2007). In contrast, AtNRT2.4 and AtNRT2.5 belong to cHATS and are expressed under very low nitrate environment or nitrogen starvation conditions (Kotur & Glass, 2015). Therefore, we speculated that the GhNRT2.1(a-e) proteins might belong to iHATS and the GhNRT2.4b, GhNRT2.5a and GhNRT2.5b proteins belong to cHATS. Although they perform different functions, their interactions are necessary for the growth of plants under a restricted nitrogen supply. In addition, the expression of GhNRT2.1(a-e) genes first increased and then decreased, which was consistent with CsNRT2 (Li et al., 2018). Thus, we speculated that plant nitrogen requirements may regulate the expression of NRT2 genes and that many plants have similar conditions. In cassava, MeNRT2.2 was strongly induced in nitrate deficiency (You et al., 2022). In barley, HvNRT2 genes also have a similar expression (Guo et al., 2020).

The potential role of GhNRT2.1e in increasing NUE

GhNRT2.1e was the most up-regulated gene at low nitrate concentration and may be a key gene regulating nitrate uptake and transport. We validated the function of GhNRT2.1e using overexpression in A. thaliana and silencing in G. hirsutum. As transgenic cotton takes longer, but Arabidopsis is a convenient model plant to study gene function, we investigated the role of GhNRT2.1e by heterologous expression in Arabidopsis (Xiao et al., 2022). The NRT2 genes were overexpressed and can promote nitrate uptake and transport, and increase plant biomass. The cassava MeNRT2.2 gene was overexpressed and increased the fresh weight of transgenic Arabidopsis (You et al., 2022). The study of the NRT2 genes in chrysanthemum demonstrated that nitrate uptake was increased when CmNRT2.4 was used in transgenic Arabidopsis and that overexpression of CmNRT2.1 also contributed to the uptake of nitrate (Gu et al., 2016). The HvNRT2.1 gene was overexpressed and increased nitrate and nitrogen content (Guo et al., 2020). The CsNRT2.4 gene was overexpressed, increasing biomass and promoting lateral root development (Zhang et al., 2021). The results of this study demonstrated that in the presence of low nitrate supply levels, transgenic Arabidopsis promotes nitrate uptake, increasing nitrate and nitrogen accumulation and ultimately contributing to biomass accumulation (Figs. 9 and 10). Therefore, GhNRT2.1e promotes nitrate uptake and transport, resulting in the better growth potential of transgenic Arabidopsis under limited nitrate conditions. The current research results on Arabidopsis overexpression lines will provide some reference for future research on transgenic cotton.

VIGS is often used to assess the function of genes, which can inhibit gene expression (Gao et al., 2011). This study revealed no significant difference between GhNRT2.1e-silenced plants and control at normal nitrate concentrations. At low nitrate concentrations, GhNRT2.1e-silenced plants had lower dry weight, nitrogen and nitrate accumulation, chlorophyll content, and nitrogen uptake and use efficiency than control plants (Figs. 11 and 12). Similar results were obtained by knocking out or silencing the NRT2 gene in other plants. For example, HATS activity was significantly reduced in AtNRT2.1-deficient mutants (Filleur et al., 2001). In cucumber, knockdown of CsNRT2.1 also inhibited nitrate uptake (Li et al., 2018). We speculate that silencing of GhNRT2.1e affects nitrate uptake by VIGS plants, thereby inhibiting nitrate and nitrogen utilization and accumulation. The reduced nitrogen supply to stems and leaves affects plant photosynthesis and biomass accumulation. Therefore, the GhNRT2.1e gene increased nitrate uptake and transport and promoted cotton growth at low nitrate concentrations.

Previous studies of Arabidopsis and rice nitrate transporter Km data demonstrated that nitrogen transporters regulate the activities of nitrogen-metabolizing enzymes (Glass et al., 2002). Nitrogen absorbed by transporters is converted into amino acids by metabolic enzymes (Iqbal et al., 2020b). Therefore, the activity of the nitrate transporter can be judged by measuring metabolic enzyme activity and amino acid content. We found that nitrogen metabolizing enzyme activity and amino acid content were higher in transgenic Arabidopsis (Fig. 10). However, nitrogen metabolizing enzyme activity and amino acid content were found to be lower in GhNRT2.1e-silenced plants (Fig. 12). It might be due to the silencing of the GhNRT2.1e gene that inhibits nitrate absorption by VIGS plants and reduces the NO3− that enters the assimilation process, thereby affecting the assimilation process and reducing metabolic enzyme activity and amino acid content. However, transgenic Arabidopsis can absorb more NO3− and promote the metabolic process, so metabolic enzyme activity and amino acid content are increased. Previous studies on other nitrogen-related genes had similar results. The CPSF30 promoted nitrate signaling, while NR activity and amino acid content were reduced in mutants (Hou et al., 2021). BnaA2.Gln1;4 was expressed in root and shoot with a high affinity for nitrogen and BnaA2.Gln1;4 overexpression increased GS and GOGAT activity (Zhou et al., 2022). FIP1 can regulate nitrate uptake and transport, and nitrogen-metabolizing enzyme activity and nitrate content were reduced in fip1 mutants (Wang et al., 2018b).

When stressed, plants generate excessive reactive oxygen species (ROS), and cells are damaged by oxidation, which may result in plant death (Caverzan, Casassola & Brammer, 2016). MDA is a product of lipid breakdown, and the degree of peroxidation in plant cells can be measured by detecting MDA content (Wang et al., 2018a). Antioxidant enzymes function in maintaining the homeostasis of plant organelles (Hasanuzzaman et al., 2020). In this research, the oxidant and antioxidant enzyme activities were similar between GhNRT2.1e-silenced plants and controls at normal nitrate concentrations. However, at low nitrate concentrations, VIGS plants had higher MDA concentrations and lower activities of SOD, POD, and CAT than control plants (Fig. 12). Similar results were found in other studies that GhNPL5 enhanced plant tolerance under nitrogen-limiting conditions but also showed lower antioxidant enzyme activity and higher oxidant content in VIGS plants (Magwanga et al., 2019). It may be because the production and elimination of ROS are in a dynamic equilibrium in plants at normal nitrate concentrations so that they do not cause damage to plant cells. Plants are subjected to abiotic stress at low nitrate concentrations, but GhNRT2 genes can carry out high-affinity nitrate transport, maintain partial nitrogen supply, and reduce plant oxidative damage. The results indicated that GhNRT2.1e could improve cotton tolerance under low nitrogen conditions.

Interactions between proteins provide essential information for characterizing cells’ biological activities and metabolic processes (Mehari et al., 2021). NAR2 is a chaperone protein and has been identified in many plants. All AtNRT2 proteins can interact with AtNAR2.1 except AtNRT2.7 (Kotur et al., 2012). AtNRT2.5 interacts with AtNAR2.1 to create a 150 kDa complex in plasma membrane to facilitate nitrate transport (Kotur & Glass, 2015). In rice, OsNRT2.1, OsNRT2.2, and OsNRT2.3a were demonstrated to interact with OsNAR2.1 by yeast two-hybrid experiments (Yan et al., 2011). In chrysanthemum, the interaction of CmNRT2.1 and CmNAR2 in vivo was confirmed by yeast two-hybrid and BiFC assays (Gu et al., 2016). In this research, we demonstrated that GhNRT2.1e interacts with GhNAR2.1 using yeast two-hybrid and LCI assays (Fig. 13). We speculated that GhNRT2.1e and GhNAR2.1 can also form a complex in the plasma membrane, and the interaction promotes nitrate uptake and transport.

Conclusions

In this work, genome-wide identification and functional characterization of GhNRT2 genes in G. hirsutum was performed. Most NRT2 proteins were predicted at the plasma membrane. The results of phylogeny, gene structure, and conserved motifs indicated that GhNRT2 proteins were conserved during evolution. Most GhNRT2 genes were found expressed in roots by RNA-seq and qRT-PCR. When 0.25 mM nitrate was resupplied, the GhNRT2 genes showed different expression patterns. GhNRT2.1e was the most highly up-regulated and essential gene involved in nitrate uptake at low concentrations. We performed functional validation of GhNRT2.1e using overexpression in A. thaliana and silencing in G. hirsutum. Arabidopsis plants overexpressing GhNRT2.1e exhibited increased biomass, nitrogen and nitrate accumulation, nitrogen uptake and utilization efficiency, nitrogen-metabolizing enzyme activity, and amino acid content at low nitrate concentrations. VIGS plants exhibited reduced biomass, chlorophyll content, nitrogen and nitrate accumulation, nitrogen uptake and use efficiency, nitrogen-metabolizing enzyme activity, amino acid content, and tolerance under low nitrate conditions. These results demonstrated that GhNRT2.1e promoted nitrate uptake, regulation of nitrogen metabolism, and biomass accumulation. We demonstrated the interaction of GhNRT2.1e with GhNAR2.1 by yeast two-hybrid and LCI assays and speculated that they work together to promote nitrate transport. This study illustrates the function of GhNRT2 genes and provides new ideas for improving NUE, increasing yield, and developing new varieties with high nitrogen utilization.

Supplemental Information

Supplemental Information 1 The primers used in this study.

Click here for additional data file.

Supplemental Information 2 The information of NRT2 genes in cotton.

Click here for additional data file.

Supplemental Information 3 Duplication information of the GhNRT2 genes.

Click here for additional data file.

Supplemental Information 4 Ka/Ks Values of all duplicated gene pairs from four Gossypium species.

Click here for additional data file.

Supplemental Information 5 Cis-element statistics of GhNRT2 genes.

Click here for additional data file.

Supplemental Information 6 The raw data of qRT-PCR, GhNRT2.1e-overexpressed Arabidopsis and GhNRT2.1e-silenced cotton.

These data were used for statistical analysis to analyze the expression patterns of GhNRT2 genes and the function of GhNRT2.1e gene.

Click here for additional data file.

Supplemental Information 7 PCR identification of GhNRT2.1e-overexpressed Arabidopsis.

Genomic DNA from three T2 generation overexpressed lines was examined by PCR. “WT” represents the wild type; “OE” represents overexpression line.

Click here for additional data file.

Additional Information and Declarations

Competing Interests

Author Contributions

Data Availability

The authors declare that they have no competing interests.

Xinmiao Zhang conceived and designed the experiments, performed the experiments, analyzed the data, prepared figures and/or tables, authored or reviewed drafts of the article, and approved the final draft.

Jiajia Feng analyzed the data, prepared figures and/or tables, authored or reviewed drafts of the article, and approved the final draft.

Ruolin Zhao performed the experiments, authored or reviewed drafts of the article, and approved the final draft.

Hailiang Cheng analyzed the data, authored or reviewed drafts of the article, and approved the final draft.

Javaria Ashraf analyzed the data, authored or reviewed drafts of the article, and approved the final draft.

Qiaolian Wang analyzed the data, authored or reviewed drafts of the article, and approved the final draft.

Limin Lv analyzed the data, authored or reviewed drafts of the article, and approved the final draft.

Youping Zhang analyzed the data, authored or reviewed drafts of the article, and approved the final draft.

Guoli Song conceived and designed the experiments, authored or reviewed drafts of the article, and approved the final draft.

Dongyun Zuo conceived and designed the experiments, authored or reviewed drafts of the article, and approved the final draft.

The following information was supplied regarding data availability:

The raw data is available in the Supplemental Files.

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
