# Peer review of "Functional characterization of the GhNRT2.1e gene reveals its significant role in improving nitrogen use efficiency in Gossypium hirsutum"

_PeerJ, doi:10.7717/peerj.15152_

## Round 0.1 · original submission · Major Revisions

Please revise the article considering the suggestions/queries of reviewers.

·

Basic reporting

I believe that this is a well-written manuscript that is a great resource for improving crop nitrogen use efficiency (NUE) and reducing the excessive application of nitrogen fertilizer. I also believe that this is an appropriate journal for this work. I have a few comments that once resolved, I believe make the manuscript suitable for acceptance.
1. A few grammatical mistakes in the text lead to minor confusion.
2. Recheck the reference section with the in-text citations, some references are missing in the reference section. Please maintain the same referencing style throughout, following the journal’s prescribed reference style. Also, try to use the latest references.
3. Figures are good however can be improved for example in figure 7 for all graphs authors can use one and shared x-axis details

Experimental design

Question 1: The introduction and M&M are too long and full of unnecessary information, please shorten them. Also, it is not necessary to address each sentence using 4-5 refs 2-3 recent ones are enough to check lines 59-68.
Question 2: my main concern is about the gene silencing assay did the authors perform PCR to validate the gene transformation?
Question 3: Tissue-specific expression pattern analysis: how authors performed it using a general promoter (35S)? Instead of RNA seq why not a specific promoter?
Question 4: The discussion should be more specific and not generalized; please provide appropriate references and improve the text. It is advisable to use different subtitles.
Question 5: what were the internal genes for DNA quality assay as well as qRT-PCR assay?
Question 6: Agrobacterium tumefaciens flower soaking method used in the overexpression assay that’s fine however how authors can be sure if the observed overexpression is not because of Agrobacterium coexistence with plant cells? Did they perform vir genes PCR?

Validity of the findings

I believe that the manuscript is a great resource for improving crop nitrogen use efficiency (NUE) and reducing the excessive application of nitrogen fertilizer.

Additional comments

The manuscript is scientifically written in a detailed manner and the research subject is attractive. However, before acceptance, the manuscript should be checked for technical errors, grammatical, and other typographical errors. The references must be rechecked.

·

Basic reporting

a) Line 49: Spelling of “Gossypium” in keywords
b) Line 59, 60: No need of adding nitrogen after “nitrate” and “ammonium”
c) Line 131: Rewrite the sentence “Download the 132 protein sequences of AtNRT2 in Arabidopsis from TAIR” to make it grammatically correct.
d) Figure 3: spelling of hirsutum in the figure description
e) Check spacing throughout the manuscript
f) Please mention software used in the figure descriptions too
g) Line 585, 613: Replace “found” with “demonstrated” or another appropriate word
h) Line 675: “Overexpression of GhNRT2.1e in Arabidopsis exhibited increased…” should be rewritten as “Arabidopsis plants overexpressing GhNRT2.1e exhibited increased…”

Experimental design

A well-structured and systematic research on the functional analysis of the gene GhNRT2.1e, which promotes nitrate uptake and transport under low nitrate conditions. My only suggestion is that it would better to use two different internal controls for qRT-PCR analysis.

Validity of the findings

No comments.

Additional comments

i applaud the authors for such interesting study on the functional analysis of the important gene GhNRT2.1e, the well-structured and systematic research and a well-written manuscript. Keep up the good work!

---

## Round 0.2 · accepted · Accept

The article is accepted as both reviewers are satisfied with the revised version.

·

Basic reporting

The authors have improved and revised the part of the manuscript based on the comments on the earlier version.

Experimental design

.

Validity of the findings

.

Additional comments

.

·

Basic reporting

All the suggestions made has been corrected in the revised manuscript.

Experimental design

no comment

Validity of the findings

No comments

Additional comments

No comments